# Clinical, Biochemical, and Sonographic Factors Influencing Performance of Parathormone Washout Measurement vs. ^99m^Tc-MIBI Scintigraphy in the Preoperative Diagnostics of Parathyroid Adenomas

**DOI:** 10.3390/jcm12124097

**Published:** 2023-06-17

**Authors:** Ewelina Szczepanek-Parulska, Dorota Filipowicz, Rafał Czepczyński, Dominika Wietrzyk, Martyna Adamska, Nadia Sawicka-Gutaj, Maja Cieślewicz, Barbara Bromińska, Piotr Stajgis, Marek Ruchała

**Affiliations:** 1Department of Endocrinology, Metabolism and Internal Medicine, Poznan University of Medical Sciences, 60-355 Poznań, Poland; dorota.filipowicz123@gmail.com (D.F.); czepczynski@ump.edu.pl (R.C.); dominikawietrzyk@gmail.com (D.W.); adamskam107@gmail.com (M.A.); nsawicka@ump.edu.pl (N.S.-G.); barbarabrominska@ump.edu.pl (B.B.); mruchala@ump.edu.pl (M.R.); 2Department of Neuroradiology, Poznan University of Medical Sciences, 60-355 Poznań, Poland; piotr.stajgis@ump.edu.pl

**Keywords:** hyperparathyroidism, ultrasound, parathormone, washout, MIBI scintiscan

## Abstract

The purpose of the study was to assess the clinical, biochemical, and sonographic factors influencing the performance of parathormone washout measurement (PTHw) vs. MIBI in the preoperative localization of parathyroid adenoma (PA). The studied group consisted of 39 patients with primary or tertiary hyperparathyroidism. The measurement of PTH concentrations was performed using an electro-chemiluminescence immunoassay. Scintigraphic localization of PA was carried out using dual-tracer planar neck scintigraphy, using 74 MBq ^99m^Tc-pertechnetate and 740 MBq of ^99m^Tc-MIBI. MIBI was unambiguously positive in 74% of patients. Among patients with negative or inconclusive MIBI, 90% had a positive PTHw result. Among patients with negative PTHw, two out of three had a positive MIBI result. The PTHw of lesions <10 mm in their largest diameter yielded positive results in 95%, compared to 75% for MIBI. For lesions ≥10 mm in largest diameter, 88% were visualised using MIBI. In conclusion, PTHw is a highly effective, easy, quick, safe, and relatively cheap procedure which might be considered for PA localisation, especially in patients with lesions presenting typical ultrasound features and a size below 10 mm. MIBI remains a useful procedure in specialized centres, particularly for patients in whom PTHw failed, larger lesions, and in cases of the ectopic location of PA.

## 1. Introduction

The precise preoperative visualization of parathyroid adenoma (PA) is mandatory for patients with primary hyperparathyroidism (PHPT) to qualify for minimally invasive parathyroidectomy [1,2]. Unfortunately, up to 20% of imaging results are inconclusive [3]. Neck ultrasonography and ^99m^Tc-MIBI scintigraphy (MIBI) are first-line imaging techniques widely used for PA localization [4]. The sensitivity and specificity of ultrasonography is estimated to be 69–90% and 90–98%, respectively. For MIBI, similarly wide ranges have been reported (68–95%) [5]. The combination of both methods shows better performance (sensitivity and specificity 90–100%), while the highest diagnostic accuracy (97%) was achieved using the parathormone (PTH) concentration measurement in the washout from the ultrasound-guided biopsy (PTHw) of the suspected lesion [6].

The aim of the study was to assess the clinical, biochemical, and sonographic factors influencing the performance of PTHw vs. MIBI in the preoperative localization of PA.

## 2. Materials and Methods

The studied group consisted of 39 patients (34 females and 5 males). Mean age of subjects was 57.4 years; min–max was 32–74 years. Patients were diagnosed with primary (36 subjects) and tertiary (3 subjects) HPT and were referred to the department of endocrinology for localization diagnostics before qualifying for parathyroidectomy from January 2018 to December 2020.

The PTHw procedure was performed with a fine needle (lumen diameter 0.5 mm), under ultrasound guidance. The aspirate was dissolved in 1 mL of 0.9% NaCl. PTHw and PTH serum (PTHs) concentration measurements were performed using the electro-chemiluminescence third-generation immunoassay, Elecsys PTH 1–84, using a cobas e 801 analyzer (serum normal range 15–57 pg/mL, upper limit of detection—ULD—2300 pg/mL). Results above the upper limit of detection were assumed as equal to the upper limit of detection.

Ultrasound examination was performed with the AIXPLORER system (Supersonic Imagine, Aix en-Provence, France). PTHw was considered positive if PTHw was ≥1.0 of PTHs.

Scintigraphic localization of PA was carried out using dual-tracer planar neck scintigraphy and a Nucline gamma camera (Mediso, Budapest, Hungary). First, planar images of the neck area were acquired 15 min after an injection of 74 MBq of ^99m^Tc-pertechnetate (^99m^Tc generator by Polatom, Otwock, Poland) to visualise tracer distribution in the thyroid gland. Subsequently, patients were injected with 740 MBq of ^99m^Tc-MIBI (PoltechMIBI, Polatom, Otwock, Poland), and anterior images of the neck were registered twice: 15 min and 120 min after injection. Any focus of increased ^99m^Tc-MIBI accumulation that did not show any ^99m^Tc-pertechnetate uptake was regarded a positive for PA.

Statistical analysis was performed with the use of Statistica 13 (StatSoft, Tulsa, OK, USA) and Microsoft Excel software. Results were presented as median with a Q1–Q3 interval. In addition to the descriptive statistics, non-parametric tests were used for comparison (Mann–Whitney U test) and correlation (Spearman test). The level of statistical significance was set at *p* < 0.05. The study was retrospective, based on the analysis of medical documentation. Thus, the requirement of approval was waived by our institutional Bioethical Review Board.

## 3. Results

The surgery was effective (PA was histopathologically confirmed and/or PTHs postoperatively normalized) in 97% of patients (38/39). Biochemical characteristics of the studied group are presented in Table 1.

### 3.1. PTHw

PTHw was positive in 92% (36/39) of subjects. In 21 cases (54%), the PTHw concentration exceeded the ULD. The median PTHw concentration was significantly different between PTHw-positive and -negative subgroups (2300 pg/mL, 25–75%:1310–2300 vs. 35.8, 25–75%:35.3–818.4 pg/mL *p* = 0.005).

### 3.2. PTHw vs. MIBI

The results of PA localization obtained using ultrasound and PTHw were in agreement with the localisation suggested by MIBI in 77% (30/39) of cases. MIBI was unambiguously positive in 74% (29/39) of cases. Among patients with negative or inconclusive MIBI (multifocal or non-specific or negative uptake), 90% (9/10) had a positive PTHw result. In MIBI-positive patients, PTHw confirmed the presence of the PA location in 93% of cases (27/29). Within the group with positive PTHw, in 25% (9/36) of subjects, MIBI was inconclusive (eventually, in 8/9 cases, biochemical normalization was achieved after the resection of the PTHw positive lesion). Among patients with negative PTHw, two out of three had positive MIBI and underwent successful surgery (Figure 1).

### 3.3. PTHw and Ultrasound Results

#### 3.3.1. Features Typical of PA

The most common PA was located in the left lower parathyroid gland (44%), followed by the right lower parathyroid gland (41%). In 85% of lesions (33/39), the ultrasound image was considered typical of PA (bean-shape, hypoechogenicity, extracapsular location on the posterior wall of the thyroid gland). Among lesions presenting typical ultrasound features of PA, PTHw was positive in 97% (compared to 67% for lesions not presenting typical features) and MIBI was positive in 73% (83% of lesions without typical features).

#### 3.3.2. Size

The median diameter of the lesion suspected of PA and selected for PTHw was 9 mm (25–75%, 7–14 mm). The PTHw of the lesions < 10 mm in their largest diameter yielded positive results in 95%, compared to 75% for MIBI. On the other hand, in the group of MIBI-negative results, 80% of them had a largest diameter < 10 mm. In lesions ≥10 mm in largest diameter, 88% were visualised on MIBI, while 87% of them had positive PTHw. In the group with negative PTHw, 66% of lesions were ≥10 mm.

#### 3.3.3. Concomitant Thyroid Diseases

Upon ultrasound examination, 59% of patients had at least one concomitant thyroid focal lesion, and 38% demonstrated features of autoimmune thyroid disease (AITD), namely, decreased echogenicity and inhomogeneous structure of the gland parenchyma. All three subjects with negative PTHw had nodular goitre. Among patients with negative PTHw, one out of three cases had AITD. Seventy percent of patients (7/10) with negative or inconclusive MIBI had concomitant thyroid focal lesions. In the ultrasound results, among cases with negative or inconclusive MIBI, AITD features were present in half of the cases (5/10).

### 3.4. Predictive Factors of PTHw Utility

The PTHs (R = 0.4, *p* = 0.01) and PTHw/PTHs concentration ratio (R = −0.4, *p* = 0.01) correlated with the maximum diameter of the lesion on ultrasound. Additionally, eGFR was significantly higher in the group with positive PTHw than those with a negative result (90.8 vs. 57.7, *p* = 0.04). In the ROC curve analysis, serum calcium level ≤ 11.2 mg/dL predicted PTHw confirmation with a sensitivity of 58.8% and specificity of 100% (*p* = 0.0059, AUC = 0.747).

## 4. Discussion

A recent meta-analysis estimated the pooled sensitivity of the PTHw as 95%, specificity equal to 83%, while positive and negative predictive values were 97% and 73%, respectively [6]. The diagnostic accuracy of PTHw evaluated by Canpolat et al. was far better (90.8%) than MIBI (67.8%) and the cytological examination (7.92%) [7]. Moreover, in a recent study, a PTHw technique demonstrated higher sensitivity (95.6%) to MIBI (52.2%), with a low complication rate [8].

The efficacy of PTHw in the proper localization of PA is higher in lesions presenting sonographic features typical of PA and a maximum lesion diameter below 10 mm. Although there is no particular threshold of size, the sensitivity of MIBI increases the PA size [9], which is consistent with our findings. PTHw has lower diagnostic potential in cases of multifocal PA or parathyroid hyperplasia (i.e., in case of MEN syndrome), as PA may mimic thyroid lesions or lymph nodes [10,11]. In our group, PTHw usefulness is also limited in patients with concomitant nodular goitre and/or AITD, in case of higher PTHs values (due to a falsely low PTHw/PTHs ratio) and chronic kidney disease (decreased eGFR). This is consistent with an observation made by Popowicz et al., who also found that the presence of nodular goitre and/or AITD exerts a negative impact on the percentage of negative results of PTHw; however, an effect was still two-times more pronounced for SPECT-CT [12,13]. Although there is still no clearly recommended cut-off for PTHw concentration, most authors agree with the need of ratios of PTHw/PTHs rather than a cut-off concentration [7].

The analysed group included three patients with tertiary hyperparathyroidism. In all of them, both PTHw and MIBI scintiscans were positive and indicated the parathyroid adenoma localisation well. However, a larger group of patients is needed to draw any conclusions on this specific subgroup of patients in terms of PTHw utility.

Compared to MIBI, PTHw is cheaper and quicker (can be performed during one visit together with the first ultrasound examination in an outpatient manner). Of note, MIBI exposes a patient to radiation (and is therefore not applicable to pregnant females and, in lactating women, requires a pause in breastfeeding); therefore, it requires access to a nuclear medicine laboratory and may provide falsely negative or inconclusive results, particularly in cases of small lesions [14,15]. On the other hand, the limitation of PTHw is that it can only be performed (especially in lesions below 10 mm) by a highly skilled physician with experience in fine-needle aspiration biopsy. The contraindications and complications are similar to those of thyroid biopsy and should not be neglected [16].

The suboptimal results of PA detection achieved using MIBI in this study may be attributed to the fact that only planar images were analysed. With technical progress, SPECT-CT, as an additional modality, has also been introduced in our centre. Moreover, it has highly improved the PA detection rate [17,18]. Unfortunately, while recruiting for the study, SPECT-CT was not routinely performed in all cases. Therefore, these data have not been included in the current study.

Interestingly, even higher values of diagnostic accuracy have been obtained with PET-CT using ^18^F-choline, even for small Pas [3,19]. In a recent meta-analysis, the sensitivity of ^18^F-choline PET was equal to 95%, with a very high PPV of 97%. The procedure is simple, involves only one tracer injection, and the image acquisition is restricted to several minutes. Its disadvantages involve the high cost and relatively low availability of the radiopharmaceutical in comparison to ^99m^Tc-MIBI. However, radionuclide imaging continues to play a pivotal role in the diagnostics of HPT. A particular advantage is its ability to detect ectopic PA [17,19].

A limitation of our study was the relatively low number of patients; however, only patients who could be surgically and histopathologically verified were included in the analysis.

## 5. Conclusions

In conclusion, PTHw is a highly effective, easy, quick, safe, and relatively cheap procedure which can be considered for PA localisation, especially in patients with lesions presenting typical ultrasound features and that are below 10 mm and are accessible for ultrasound-guided biopsy. MIBI remains a useful procedure in specialized centres, particularly for patients for whom PTHw failed to identify a source of PTH overproduction, larger lesions (>10 mm), and in cases where there is a suspicion of the ectopic location of PA. Therefore, an adequate patient profile needs to be considered for each method. In order to obtain the best diagnostic results, PTHw and MIBI scintigraphy should be applied as complementary techniques. Further studies are required to increase the evidence.

## Figures and Tables

**Figure 1 jcm-12-04097-f001:**
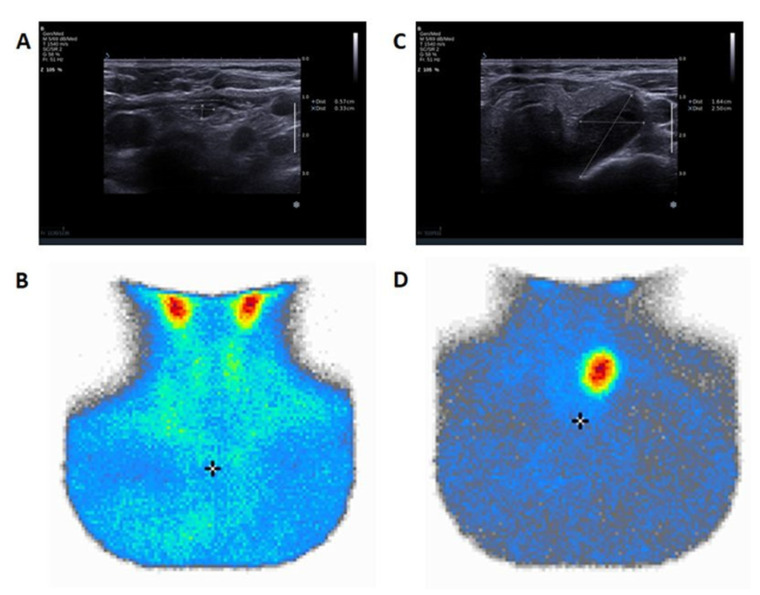
Two examples of patients with hyperparathyroidism due to parathyroid adenoma and discordant results of ultrasound examination and MIBI scintiscan: (1) ultrasound examination of a 60-year-old patient with left lower parathyroid adenoma (6 × 3 × 6 mm in size. (**A**) identified using positive PTH washout and not presenting increased radioisotope uptake in MIBI scintisccan ) (**B**); (2) ultrasound examination of a 73-year-old patient with an upper left parathyroid adenoma sized 16 × 25 × 28 mm. (**C**), with confirmed increased radioisotope uptake in MIBI scintiscan (**D**) but with a negative PTH washout result. The “*” in the image indicates the freeze mode. The “+” in the image stands for the jugular notch of the sternum.

**Table 1 jcm-12-04097-t001:** The studied group’s biochemical characteristics.

Biochemical Parameters	Median (Q1–Q3)	Reference Range (Unit)
PTHw (+) < ULD	1188 (803–1493)	-
serum PTH (PTHs)	129 (101–192)	15–57 (pg/mL)
total-calcium (Ca)	11.2 (10.7–11.7)	8.80–10.20 (mg/dL)
ionized-calcium (Ca^2+^)	6 (5.7–6.6)	4.20–5.20 (mg/dL)
inorganic phosphates (Pi)	2.6 (2.3–2.8)	2.70–4.50 (mg/dL)
vitamin D (25-OH-D)	19.5 (13–25)	30–80 (ng/mL)
eGFR by MDRD	84 (76–105)	>60 (ml/min/m^2^)
24 h-Ca_u_	309 (256–380)	100–320 (mg/24 h)
24 h-P_u_	0.6 (0.5–0.8)	0.4–1.3 (g/24 h)
ALP	124 (75–187)	35–105 (U/L)

ULD—upper limit of detection, eGFR—estimated Glomerular Filtration Rate, MDRD—Modification of Diet in Renal Disease, 24 h-Ca_u_—calcium in 24-h urine collection, 24 h-P_u_—phosphates in 24-h urine collection, ALP—alkaline phosphatase, PTHw (+) < ULD—positive PTH washout after exclusion of results equal or higher to ULD (2300 pg/mL).

## Data Availability

The data presented in this study are available upon request from the corresponding author.

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
