# Peer review of "Clinical, Biochemical, and Sonographic Factors Influencing Performance of Parathormone Washout Measurement vs. 99mTc-MIBI Scintigraphy in the Preoperative Diagnostics of Parathyroid Adenomas"

_jcm, 2023, doi:10.3390/jcm12124097_

Round 1
Reviewer 1 Report
The determination of PTH in the lavage is undoubtedly a useful technique in the diagnosis of localization of primary hyperparathyroidism. Perhaps the main indication would be in those cases in which localization techniques with MIBI, performed under the best conditions (including SPECT/CT) are not diagnostic. In the present study, better results were obtained with PTHw in lesions smaller than 10 mm, which are those below the resolution limit of scintigraphy, and above all without the use of SPECT/CT.
Reference is made to 3 patients with tertiary hyperparathyroidism, a totally insufficient number to draw conclusions.
The limitations of the study are well referenced.
The conclusion is too blunt. Lesions smaller than 10 mm are those that require greater experience of the physician to perform ultrasound, so the results in routine practice would not be as good.
The most realistic conclusion would be that PTHw and scintigraphic explorations are complementary, there is an adequate patient profile for each of them and the important thing is the adequate selection of patients. More studies are needed to increase the evidence
Author Response
The determination of PTH in the lavage is undoubtedly a useful technique in the diagnosis of localization of primary hyperparathyroidism. Perhaps the main indication would be in those cases in which localization techniques with MIBI, performed under the best conditions (including SPECT/CT) are not diagnostic. In the present study, better results were obtained with PTHw in lesions smaller than 10 mm, which are those below the resolution limit of scintigraphy, and above all without the use of SPECT/CT.
Reference is made to 3 patients with tertiary hyperparathyroidism, a totally insufficient number to draw conclusions.
The limitations of the study are well referenced.
The conclusion is too blunt. Lesions smaller than 10 mm are those that require greater experience of the physician to perform ultrasound, so the results in routine practice would not be as good.
The most realistic conclusion would be that PTHw and scintigraphic explorations are complementary, there is an adequate patient profile for each of them and the important thing is the adequate selection of patients. More studies are needed to increase the evidence
Re: Thank you very much for your suggestions. In order to improve the sound and clarity of our conclusions, we have added three sentences to the previous version.
In conclusion, PTHw is a highly effective, easy, quick, safe and relatively cheap procedure, which might be considered for PA localisation, especially in patients with lesions presenting typical ultrasound features and those of size below 10 mm that are accessible for ultrasound-guided biopsy. MIBI remains the useful procedure in specialized centres, particularly for patients in whom PTHw failed to identify source of PTH overproduction, larger lesions (>10 mm) and in case of suspicion of ectopic location of PA. Therefore an adequate patient profile needs to be considered for each method. In order to obtain best diagnostic results PTHw and MIBI scintigraphy should be applied as complementary techniques. Further studies are required to increase the evidence.
Regarding the patients with tertiary hyperparathyroidism: our subgroup of patients with tertiary hyperparathyroidism is very small, thus drawing conclusions form these results is unwarranted. Therefore, in our conclusions we do not mentioned any specific findings and recommendations regarding tertiary hyperparathyroidism. However, we added to the manuscript an information on the results of these three patients. In all three patients both PTHw and MIBI scintiscan were positive and indicated well the parathyroid adenoma localisation. The clinical differences exist, however were not the subject of the present analysis, the same we did not analyze the clinical profile of patients with primary hyperparathyroidism. In terms of biochemical finding, patients with tertiary hyperparathyroidism tend to present higher PTH serum concentration, thus it is possible that the ratio between serum and washout fluid is relatively lower and in some may be considered falsely negative, depending on the cut off value of ratio. In our study the cut off ratio was 1, thus these patients fulfilled the criteria of positive PTHw.
Reviewer 2 Report
The authors assessed the clinical, biochemical and sonographic factors influencing performance of parathormone washout measurement (PTHw) vs. MIBI in the preoperative localization of parathyroid adenoma. The topic is of interest and the manuscript is well written. However, a few remarks should be addressed:
1. In Figure 1, the author demonstrated two examples of patients with hyperparathyroidism due to parathyroid adenoma and discordant results of ultrasound examination and MIBI scintiscan. Could you explain the reasons which caused the discordant results?
2. As the authors mentioned, the accuracy of PTHw depends on the experience of physician. Were the PTHw procedure in this study performed with one or more experienced physician? How to reduce the bias between different physicians?
3. This study recruited 36 subjects with primary hyperparathyroidism and 3 subjects with tertiary hyperparathyroidism. Could you illustrate the differences between primary and tertiary hyperparathyroidism in clinical, biochemical and sonographic factors?
Author Response
Thank you very much for your suggestions. We enclose answers to the suggestions below. In order to improve the sound and clarity of our conclusions, we have added some amendments to the previous version.
The authors assessed the clinical, biochemical and sonographic factors influencing performance of parathormone washout measurement (PTHw) vs. MIBI in the preoperative localization of parathyroid adenoma. The topic is of interest and the manuscript is well written. However, a few remarks should be addressed:
1. In Figure 1, the author demonstrated two examples of patients with hyperparathyroidism due to parathyroid adenoma and discordant results of ultrasound examination and MIBI scintiscan. Could you explain the reasons which caused the discordant results?
Re:
Discordant results illustrate the conclusion that PTHw and MIBI scyntygraphy should be considered as compementary techniques and both performed in each patient.
The reason why we obtained the discordant results – in the first patient - is probably due to the small size of the parathyroid adenoma, which is below 1 cm and therefore below the resolution and limit of detection by MIBI scintiscan. However, still could be reached by biopsy and the material allowed of assessment of PTH concentration in the washout fluid, and confirm the localization of parathyroid adenoma. In case of the second patient, the lesion was large and therefore could be detected by MIBI scintiscan. However, it is difficult to tell why PTH concentration was relatively low in the washout fluid. We may speculate that larger lesion may be more stiff, more degenerated, contain fibrotic or necrotic changes, thus the collection of the material in large portion may be difficult and therefore the result of PTHw measurement might be falsely negative. The other issue is the fact that in larger lesions, usually the serum PTH concentration is very high (even above the upper limit of detection, thus we needed to assume that the concentration was equal to upper limit of detection – we added this explanation to the methods section), thus the concentration of the PTH in washout fluid obtained should also be very high to achieve the adequate ratio to consider the PTHw/PTHs ration positive.
,,Two examples of patients with hyperparathyroidism due to parathyroid adenoma and discordant results of ultrasound examination and MIBI scintiscan:
1) 60-year-old patient with left lower parathyroid adenoma sized 6x3x6 mm on ultrasound examination (A), identified by positive PTH washout and not presenting increased radioisotope uptake in MIBI scintisccan (B)
- PTH wash out is especially useful in patients with lesions presenting typical ultrasound features and those of size below 10 mm that are accessible for ultrasound-guided biopsy. This was observed in our study as the PTHw of the lesions <10 mm in its largest diameter, yielded positive results in 95%, compared to 75% for MIBI.
2) 73-year-old patient with upper left parathyroid adenoma sized 16x25x28 mm on ultrasound examination (C), with confirmed increased radioisotope uptake in MIBI scintiscan (D), but with a negative PTH washout result.
This example is consistent with our findings, as we observed that: in lesions ≥10 mm in largest diameter, 88% were visualised on MIBI, while 87% of them had positive PTHw. In the group with negative PTHw, 66% of lesions were ≥10 mm. MIBI remains the useful procedure particularly for patients in whom PTHw failed to identify source of PTH overproduction, larger lesions (>10 mm) and in case of suspicion of ectopic location of PA.
2. As the authors mentioned, the accuracy of PTHw depends on the experience of physician. Were the PTHw procedure in this study performed with one or more experienced physician? How to reduce the bias between different physicians?
Re:
All the examinations were performed by the same experienced specialist to reduce the possibility of the influence of different level of experience on the results. As it was mentioned in the paper, it is important that the PTH washout is performed by the person well skilled in ultrasound and FNAB. Therefore, the PTH washout should be prformed in referenced center, to avoid misinterpretation.
3. This study recruited 36 subjects with primary hyperparathyroidism and 3 subjects with tertiary hyperparathyroidism. Could you illustrate the differences between primary and tertiary hyperparathyroidism in clinical, biochemical and sonographic factors?
Re:
Our subgroup of patients with tertiary hyperparathyroidism is very small, thus drawing conclusions form these results is unwarranted. Therefore, in our conclusions we do not mentioned any specific findings and recommendations regarding tertiary hyperparathyroidism. However, we added to the manuscript an information on the results of these three patients. In all three patients both PTHw and MIBI scintiscan were positive and indicated well the parathyroid adenoma localisation. The clinical differences exist, however were not the subject of the present analysis, the same we did not analyze the clinical profile of patients with primary hyperparathyroidism. In terms of biochemical finding, patients with tertiary hyperparathyroidism tend to present high PTH serum concentration, thus it is possible that the ratio between serum and washout fluid is relatively lower and in some may be considered falsely negative, depending on the cut off value of ratio. In our study the cut off ratio was 1, thus these patients fulfilled the criteria of positive PTHw.